# Evaluation of the antidiarrheal activity of dichloromethane-methanol crude extract of the aerial parts of *Croton kinondoensis* (Euphorbiaceae) in Mice

**Abdullahi Ibrahim Osman Noor** *, **Ermias Mergai Terefe**

Department of Pharmacology and Pharmacognosy, School of Pharmacy and Health Sciences, United States International University – Africa, Nairobi, Kenya

* aiosman@usiu.ac.ke

## Abstract

### Background

Diarrhea remains a major global health concern, particularly in developing countries. Resistance to conventional antibiotics underscores the need for effective, plant-based alternatives. *Croton kinondoensis*, a traditional Kenyan remedy for gastrointestinal ailments, has yet to be scientifically validated. This study assessed its antidiarrheal and antimotility effects *in vivo* mouse models.

### Methods

Leaf extracts were prepared using dichloromethane/methanol (1:1). Phytochemical screening confirmed the presence of flavonoids, tannins, alkaloids and phenolics. Acute toxicity was assessed via OECD guideline 425. Antidiarrheal activity was tested using a castor oil-induced diarrhea model, and gastrointestinal motility was evaluated using the charcoal meal test. Mice were divided into five groups (n = 6): negative control (distilled water), positive control (loperamide 3 mg/kg), and three test groups receiving *C. kinondoensis* extract at doses of 100, 200, and 400 mg/kg). Statistical significance was determined using ANOVA and Tukey's post hoc test to determine significance (p < 0.05).

### Results

The extract showed no toxicity at 2000 mg/kg. In the castor oil-induced diarrhea model, the *C. kinondoensis* extract at 400 mg/kg inhibited diarrhea by 25.57% ($p < 0.05$), compared to 71.43% inhibition by loperamide ($p < 0.001$) e. The charcoal meal transit test, 400 mg/kg of the extract reduced intestinal transit by 27.98% ($p < 0.01$), compared to 33.43% by loperamide ($p < 0.001$.

**Data availability statement:** All relevant data are within the paper and its Supporting Information files.

**Funding:** The author(s) received no specific funding for this work.

**Competing interests:** The authors have declared that no competing interests exist.

## Conclusion

*C. kinondoensis* exhibits dose-dependent antidiarrheal and antimotility effects, supporting its traditional use. Although its efficacy was lower than loperamide, the extract demonstrated significant potential as a natural remedy for diarrhea.

## Introduction

Diarrhea remains a major global health challenge, particularly in developing countries [1], where it is a leading cause of morbidity and mortality. The World Health Organization (WHO) estimates that diarrhea-related diseases account for approximately 525,000 child deaths annually, with rotavirus being one of the primary causative agents. Despite advancements in medical treatments, including antibiotics, probiotics, and rehydration therapy, the increasing emergence of antibiotic resistance and the financial burden associated with conventional treatments necessitate the exploration of alternative therapies [2].

Traditional medicinal plants have long been used to manage gastrointestinal disorders, including diarrhea and abdominal cramps. One such plant is *Croton kinondoensis*, an endemic species found in the coastal regions of Kenya. The Euphorbiaceae family, to which *Croton kinondoensis* belongs, includes several species known for their pharmacological properties, including antimicrobial, antispasmodic, and anti-inflammatory effects. However, despite its widespread traditional use, scientific validation of *C. kinondoensis*'s efficacy and safety remains limited.

Gastrointestinal disorders such as diarrhea, irritable bowel syndrome (IBS), and gastroenteritis significantly impact patients' quality of life, often leading to complications such as dehydration and malnutrition. The need for effective, affordable, and accessible treatments is particularly urgent in resource-limited settings where conventional medications may not be readily available. Several Croton species have demonstrated potential antidiarrheal and antispasmodic properties, making *C. kinondoensis* a promising candidate for further investigation [3].

Beyond the immediate health implications, diarrhea contributes to economic strain in affected communities. Frequent episodes of diarrhea lead to increased healthcare costs, lost productivity, and school absenteeism in children [4]. Moreover, antibiotic overuse in the treatment of infectious diarrhea has exacerbated global concerns regarding antimicrobial resistance, further emphasizing the urgency for plant-derived therapeutic alternatives. Medicinal plants like *C. kinondoensis* offer a sustainable, locally available, and cost-effective alternative for managing diarrhea and gastrointestinal disorders.

The ethnobotanical knowledge of *C. kinondoensis* is deeply embedded in indigenous medical practices, with local healers often prescribing its extracts for digestive ailments. However, due to the lack of standardized dosing and scientific validation, the plant's therapeutic potential remains largely unexplored in modern medicine. Rigorous pharmacological evaluation is essential to bridge the gap between traditional use and clinical application.

This study aims to evaluate the antidiarrheal and antispasmodic properties of *C. kinondoensis* using in vivo models. By assessing its pharmacological effects on castor oil-induced diarrhea and gastrointestinal motility, this research seeks to provide scientific evidence supporting its traditional medicinal use. Furthermore, the study will analyze the plant's phytochemical composition to identify bioactive compounds responsible for its observed effects. Understanding the mechanisms of action of these bioactive compounds can further enhance their potential application in developing standardized herbal formulations.

The findings of this study could contribute significantly to the growing body of knowledge on plant-based therapies for gastrointestinal disorders. By validating the therapeutic potential of *C. kinondoensis*, this research may pave the way for the development of novel, plant-derived antidiarrheal medications, ultimately improving healthcare accessibility and reducing dependency on conventional pharmaceutical treatments. Future investigations may also explore the synergistic effects of *C. kinondoensis* with existing antidiarrheal agents, potentially enhancing treatment efficacy and reducing drug-related adverse effects.

## Methodology

This study investigated the antidiarrheal activity of *Croton kinondoensis* using Swiss albino mice and established pharmacological models. A systematic approach was employed, including plant extraction, phytochemical screening, acute toxicity testing, and in vivo experiments.

### Animal model

Swiss albino mice (10–12 weeks old, weighing 25–37g) were obtained from the Pharmacology Department at the United States International University – Africa. The mice were maintained in a controlled environment with a 12-hour light/dark cycle and provided standard pellet diets and water ad libitum. Acclimatization was conducted for one week before experimentation. All protocols adhered to international guidelines for laboratory animal care [5].

### Plant material collection and extraction

Fresh leaves of *Croton kinondoensis* were collected from Kaya Kinondo Forest, Kenya. The leaves were washed, air-dried at room temperature, and ground into fine powder. A cold maceration process was employed using a 1:1 mixture of dichloromethane (DCM)/methanol (MeOH).

The 1:1 dichloromethane-methanol (DCM:MeOH) solvent system was selected to facilitate the extraction of both polar and moderately non-polar phytoconstituents. Dichloromethane aids in solubilizing less polar compounds such as terpenoids and steroids, while methanol efficiently extracts polar compounds including flavonoids, alkaloids, and phenolics. This mixed solvent approach has been shown to enhance the yield and diversity of secondary metabolites from medicinal plants, particularly when screening for broad pharmacological activity such as antidiarrheal effects [6].

The extract was filtered, re-macerated, and evaporated using a hot oven to remove solvents. The final yield was stored in an airtight container [6–8].

### Phytochemical screening

Phytochemical screening was performed to detect the presence of various bioactive compounds in *Croton kinondoensis* extract using standard qualitative procedures [9–11]. The tests conducted included flavonoids, phenolic compounds, alkaloids, and tannins, each following well-established protocols.

Test for Flavonoids (Shinoda Test) A piece of magnesium ribbon was added to 2 mL of an aqueous crude extract in a test tube, followed by the dropwise addition of concentrated hydrochloric acid. The formation of a pink or reddish color indicated the presence of flavonoids. This test relies on the reduction of flavonoids by magnesium in an acidic medium, leading to characteristic color changes [9]

Test for Phenolic Compounds (Ferric Chloride Test) Approximately 50 mg of the extract was dissolved in 5 mL of distilled water. A few drops of a neutral 5% ferric chloride solution were added to the mixture. The appearance of a dark green, blue, or black coloration indicated the presence of phenolic compounds. The principle behind this test is the formation of colored complexes between phenols and ferric chloride [11]

Test for Alkaloids (Mayer's Test) To determine the presence of alkaloids, 1 mL of the extract was treated with a few drops of Mayer's reagent (potassium mercuric iodide solution). The formation of a yellowish or white precipitate confirmed the presence of alkaloids. This reaction occurs due to the precipitation of alkaloidal bases with Mayer's reagent, forming insoluble complexes [10]

Test for Tannins (Ferric Chloride Test) Approximately 0.1 g of the extract was boiled in 2 mL of dimethyl sulfoxide (DMSO) for 5 minutes. After cooling, the mixture was filtered, and a few drops of 0.1% ferric chloride solution were added. The development of a brownish-green or blue-black coloration indicated the presence of tannins. This reaction is based on the formation of ferric tannate complexes, which produce distinct color changes [11].

Test for Saponins (Foam Test and Froth Test): *Foam Test:* About 1 mL of extract was shaken vigorously with 5 mL of distilled water. Persistent foam for 10 minutes indicated the presence of saponins. *Froth Test:* About 1 mL of extract was diluted with distilled water to 10 mL and shaken. Froth formation persisting for 15 minutes or more was considered a positive test for saponins.

**Test for Steroids:** Two standard methods were used to detect the presence of steroids: Two standard methods were employed to detect the presence of steroids: the Liebermann-Burchard test and the Salkowski test, as described by Harborne [12] and Trease & Evans [13]. The Liebermann-Burchard test and the Salkowski test. **Liebermann-Burchard Test**: 2 mL of the extract was mixed with 2 mL of acetic anhydride. Then, 2 mL of concentrated sulfuric acid was carefully added down the side of the test tube. The formation of a bluish-green color indicated the presence of steroids. **Salkowski Test:** 2 mL of the extract was mixed with 2 mL of chloroform, followed by the addition of 2 mL of concentrated sulfuric acid along the wall of the test tube. A red coloration at the interface was considered a positive indication of steroids.

Test for Glycosides: Two tests were conducted to detect cardiac and general glycosides as described by Harborne [12] and Trease & Evans [13]. **Keller-Kiliani Test (for Cardiac Glycosides)**: To 1 mL of the extract, 2 mL of glacial acetic acid containing a trace amount of ferric chloride was added. This was followed by carefully layering 1 mL of concentrated sulfuric acid. A reddish-brown ring at the interface indicated the presence of cardiac glycosides. **Legal's Test (for General Glycosides):** The extract was treated with sodium nitroprusside and sodium hydroxide. The development of a pink to red color indicated a positive test for glycosides.

These qualitative tests provided preliminary confirmation of key phytochemical constituents present in *Croton kinondoensis*, supporting further pharmacological evaluation of its therapeutic potential.

## Acute toxicity test

The acute toxicity study was conducted following Organization for Economic Cooperation and Development (OECD) guideline 425 [14] to evaluate the safety profile of *Croton kinondoensis* extract in Swiss albino mice. Five female mice (10–12 weeks old, 25–30 g) were used for the experiment. After a 3-hour fasting period, a single oral dose of 2000 mg/kg of the extract was administered to one mouse via oral gavage, followed by close monitoring for 14 days [14]. The study was continued with four additional mice, and observations were made for behavioral, neurological, autonomic, and gastrointestinal toxicity signs. Parameters such as tremors, respiratory distress, loss of appetite, diarrhea, and weight loss were assessed. In case of mortality, a gross necropsy would have been performed to examine organ damage.

The study complied with ethical guidelines, with approval from the United States International University – Africa and NACOSTI (NACOSTI/P/23/30200). These findings provide preliminary evidence of the extract's safety, supporting its potential for further pharmacological studies.

## Ethical considerations

The study was conducted following the National Research Council (NRC) guidelines for the humane treatment of laboratory animals. Efforts were made to minimize animal suffering, and humane endpoints were established. The experiment was approved by the Institutional Animal Care and Use Committee (IACUC) of United States International University – Africa and the National Commission for Science, Technology & Innovation (NACOSTI) under reference number NACOSTI/P/23/30200.

## Experimental grouping and dosing

- Each group consisted of six mice (n = 6), and treatments were administered via oral gavage once as follows: Group I (Negative Control): 0.5 mL Distilled water.

- Group II (Positive Control): Loperamide 3 mg/kg

- Group III (CK100): *C. kinondoensis* extract 100 mg/kg

- Group IV (CK200): *C. kinondoensis* extract 200 mg/kg

- Group V (CK400): *C. kinondoensis* extract 400 mg/kg

## Antidiarrheal activity testing

**Castor oil-induced diarrhea model.** The castor oil-induced diarrhea model was used to evaluate the antidiarrheal activity of *Croton kinondoensis* extract in Swiss albino mice [15].

Mice were fasted for 18 hours with access to water only to ensure uniformity in food digestion and absorption. They were then randomly divided into five groups (n = 6), consisting of a negative control group (distilled water), a positive control group (loperamide 3 mg/kg), and three test groups (100 mg/kg, 200 mg/kg, and 400 mg/kg of the extract). The treatments were administered orally using an oral gavage, ensuring precise dosing. One hour after receiving their respective treatments, all mice were administered 0.5 mL of castor oil orally to induce diarrhea.

After castor oil administration, each mouse was individually placed in separate metabolic cages lined with white paper to facilitate clear observation and quantification of fecal output. The onset of diarrhea was defined as the time to the first unformed (wet) stool. The number of wet and dry fecal pellets per animal was counted over 4 hours. The percentage inhibition of diarrhea was calculated using the following formula:

$$\% \ \textbf{\textit{inhibiton}} \ = \frac{\textbf{\textit{WFC}} - \ \textbf{\textit{WFT}}}{\textbf{\textit{WFC}}} \ \textbf{\textit{X}} \ 100$$

(1)

Where **WFC** = Wet feces in the control group; **WFT** = Wet feces in the test group
Where:

- **WFC** = Wet feces in the control group

- **WFT** = Wet feces in the test group

Additionally, the **total fecal output inhibition** was determined using:

$$\% \ \textbf{\textit{ITFO}} = \frac{\textbf{\textit{Average no. of total feacal output of}} \ (\textbf{\textit{Control}} - \textbf{\textit{Test}})}{\textbf{\textit{Average of total fecal output of control}}} \ \textbf{\textit{X}} \ 100$$

(2)

Where:

- **ITFO** = Inhibition of total fecal output

Observations were made throughout the experiment to assess changes in defecation frequency, and stool consistency. Loperamide, a standard antidiarrheal drug, was used as a positive control to compare the effectiveness of the extract. Mice receiving the extract were evaluated to determine whether it could reduce stool frequency and water content, indicating its potential antidiarrheal activity.

At the end of the 4-hour observation period, mice were carefully examined. The study was conducted in compliance with ethical standards, with strict monitoring to minimize animal distress. The results were then statistically analyzed to determine whether the extract exhibited significant antidiarrheal activity compared to the control groups.

To facilitate interpretation of the extract's antidiarrheal efficacy, a standard classification system was adopted from previous studies. In this framework, percentage inhibition of diarrhea is categorized as follows: inhibition less than 20% is considered mild, 20–40% moderate, 40–60% strong, and over 60% very strong [15,16]. This scale was applied to interpret the dose-dependent effects observed in the treated groups.

**Antimotility test...** The antimotility test was conducted to evaluate the effect of *Croton kinondoensis* extract on gastrointestinal motility using the charcoal meal method [17]. This model is commonly used to determine the effect of test substances on intestinal transit, as it assesses the distance travelled by a charcoal marker through the small intestine, reflecting changes in peristaltic movement.

Mice were fasted for 18 hours with free access to water to standardize gastrointestinal conditions before the experiment. They were randomly divided into five groups (n = 6), consisting of a negative control group (distilled water), a positive control group (loperamide 3 mg/kg), and three test groups (100 mg/kg, 200 mg/kg, and 400 mg/kg of the extract). Treatments were administered orally using an oral gavage, ensuring accurate dosing.

Thirty minutes after treatment administration, all mice received 0.5 mL of castor oil orally to stimulate intestinal motility. Castor oil, through its active metabolite ricinoleic acid, induces smooth muscle contractions and promotes gastrointestinal propulsion, making it a reliable agent for evaluating antimotility effects.

Another 30 minutes post-castor oil administration, a 0.5 mL charcoal meal (a suspension of activated charcoal in a 5% gum acacia solution) was administered orally. The charcoal meal serves as a marker to visually track intestinal movement. After 30 minutes, mice were euthanized using phenobarbital (45 mg/kg) via intraperitoneal injection, followed by cervical dislocation to ensure humane sacrifice.

Immediately after euthanasia, the abdominal cavity was carefully opened, and the entire small intestine was removed, extending from the pyloric junction to the cecum. The total length of the intestine was measured, along with the distance travelled by the charcoal marker. Measurements were obtained using a calibrated ruler, ensuring accuracy. The peristalsis index (PI) and percentage inhibition of intestinal transit were calculated using the following formulas [18]:

$$\% \, \textbf{PI} = \frac{\textbf{\textit{The distance moved by the charcoal marker}}}{\textbf{\textit{The total length of the intestine}}} \, \textbf{\textit{X}} \, 100 \tag{3}$$

$$\% \, \textbf{\textit{inhibition of intestine}} = \frac{\textbf{\textit{PI of (Control − Test group)}}}{\textbf{\textit{PI of Control}}} \, \textbf{\textit{X}} \, 100 \tag{4}$$

Where:

- **PI** = Peristalsis index
- **Control group** = Mice receiving distilled water
- **Test groups** = Mice receiving different doses of the extract

Mice treated with the negative control (distilled water) were expected to show maximum charcoal propulsion, while those receiving loperamide (positive control) were anticipated to exhibit reduced motility, as loperamide slows intestinal transit through opioid receptor-mediated inhibition of peristalsis. The effectiveness of *C. kinondoensis* extract in inhibiting peristalsis was compared to loperamide, with a higher percentage of inhibition suggesting a stronger antimotility effect.

**Statistical analysis.** Data were analyzed using SPSS Version 29 at a 95% confidence interval. Results were expressed as mean ± standard deviation (SD). One-way ANOVA followed by Tukey's post hoc test was conducted to determine statistical significance ($p < 0.05$ considered significant).

## Results

### Phytochemical screening results

Phytochemical analysis of the *Croton kinondoensis* leaf extract revealed the presence of several bioactive constituents with potential pharmacological relevance. Flavonoids, phenolic compounds, alkaloids, tannins, saponins, and reducing sugars were detected, while glycosides, anthraquinones, and typical sterols were absent.

The presence of flavonoids and phenolic compounds suggests possible antioxidant and anti-inflammatory activities, which could support intestinal mucosal protection and reduce inflammation. Alkaloids were also detected and are known for their antimicrobial and spasmolytic effects, which may aid in modulating gut motility and managing diarrhea.

Tannins, identified in the extract, possess astringent properties that can reduce intestinal secretions and enhance water reabsorption. Saponins were present, albeit in lower concentrations, and their surfactant properties may contribute to membrane stabilization and antimicrobial activity. Reducing sugars were also detected and could influence the pharmacokinetics or synergistic activity of other constituents.

Steroidal compounds were indicated by a positive Salkowski test, despite a negative Liebermann-Burchard result, suggesting the presence of triterpenoid-like structures rather than classic sterols. These compounds are often associated with anti-inflammatory and smooth muscle relaxant effects, which may support the extract's antidiarrheal properties.

Glycosides and anthraquinones were absent, suggesting that the extract's mechanism does not rely on glycosidic modulation or stimulant laxative effects.

The detailed results of the phytochemical screening are summarized in Table 1 below:

These findings suggest that the bioactive constituents present in *C. kinondoensis* extract could contribute to its antidiarrheal and antispasmodic properties, supporting its traditional use in managing diarrhea and gastrointestinal disorders. Further investigation, including quantitative analysis and mechanistic studies, is necessary to confirm their pharmacological significance.

### Acute toxicity test

The acute toxicity test of *Croton kinondoensis* leaf extract was conducted following OECD guideline 425, where Swiss albino mice were administered a single high dose of 2000 mg/kg via oral gavage. The mice were observed for any signs of toxicity, behavioral changes, or mortality over an initial 24-hour period, followed by continuous monitoring for 14 days.

No mortality or significant behavioral abnormalities were observed in any of the test animals during the entire observation period. The mice remained active, with normal food and water intake, and did not exhibit signs of lethargy, tremors, convulsions, salivation, diarrhea, or respiratory distress.

Furthermore, no changes in body weight, grooming behavior, or general physiological activities were recorded throughout the 14-day period, indicating that the extract did not induce any observable acute toxic effects at the administered dose.

The results indicated that no mortality or severe toxicity signs were observed at the tested dose, suggesting that the extract is safe up to 2000 mg/kg. Since no adverse effects were recorded, a lower dose (300 mg/kg) was not tested, and the LD50 was estimated to be greater than 2000 mg/kg, indicating a wide safety margin. However, further chronic toxicity studies and histopathological evaluations are recommended to confirm its long-term safety profile.

**Table 1. Phytochemical Screening of *Croton kinondoensis* Leaf Extract.**

| Phytochemical Test | Target Compound | Observed Result | Presence (+/-) |
|---|---|---|---|
| Shinoda Test | Flavonoids | Pink coloration | + |
| Ferric Chloride Test | Phenolic Compounds | Dark green coloration | + |
| Mayer's Test | Alkaloids | Yellowish-white precipitate | + |
| Froth Test | Saponins | Low froth formation | − |
| Ferric Chloride Test | Tannins | Brownish-green coloration | + |
| Liebermann-Burchard Test | Steroids | No color change | − |
| Salkowski Test | Steroids | Reddish-brown ring at interface | + |
| Borntrager's Test | Anthraquinones | Pink to red coloration | − |
| Benedict's Test | Reducing Sugars | Green, yellow, or red precipitate | + |
| Foam Test | Saponins | Stable froth formation | + |
| Keller-Kiliani Test | Cardiac Glycosides | No reddish-brown ring | − |
| Legal's Test | Glycosides | No color change | − |

**Key:** (+) = Present, (-) = Absent

## Castor oil-induced diarrhea model

The castor oil-induced diarrhea model was used to assess the antidiarrheal activity of *Croton kinondoensis* leaf extract in Swiss albino mice. The onset of diarrhea, the total number of wet feces, and the total fecal output were recorded over 4 hours following the administration of 0.5 mL of castor oil. The effect of the extract at different doses (100 mg/kg, 200 mg/kg, and 400 mg/kg) was compared with the negative control (distilled water) and the positive control (loperamide 3 mg/kg).

Mice in the negative control group exhibited continuous diarrhea, producing a high number of wet fecal pellets. In contrast, groups treated with *Croton kinondoensis* extract showed a dose-dependent reduction in the number of wet feces and total fecal output. The highest dose (400 mg/kg) demonstrated the most significant reduction, comparable to the loperamide-treated group as highlighted in Table 2.

The onset of diarrhea was delayed in a dose-dependent manner by the extract. At 400 mg/kg, the onset was delayed by 42.3 minutes compared to 19.7 minutes in the negative control ($p < 0.05$). Total fecal output was also reduced significantly at higher doses of the extract compared to the control group.

The castor oil-induced diarrhea model confirmed the antidiarrheal activity of *Croton kinondoensis* extract, demonstrating a dose-dependent reduction in both defecation frequency and wet fecal output. Mice in the negative control group

**Table 2. Effect of *Croton kinondoensis* extract on castor oil-induced diarrhea in mice.**

| Group | Onset of Diarrhea (min) | Total Number of DF | % Inhibition of Defecation | Total Number of Wet Feces | % Inhibition of Diarrhea | Total Fecal Output (DF+WF) |
|---|---|---|---|---|---|---|
| NC | 19.7±2.1 | 1.33±0.52 | – | 3.5±1.1 | – | 4.83±1.62 |
| PC | 51.6±3.4[a3] | 0.33±0.52 | 72.41 | 1±0.63[a3c2d1] | 71.43 | 1.33±0.83[a3] |
| CK100 | 27.4±2.8[b1] | 1.33±0.51 | 10.34 | 3±0.89[b2] | 14.29 | 4.33±1.40[b2] |
| CK200 | 35.2±3.1[b1] | 1.2±1.2 | 17.24 | 2.8±1.2[b1] | 19.05 | 4.00±1.74[b1] |
| CK400 | 42.3±2.6[a2] | 1.3±0.53 | 20.69 | 2.5±0.55 | 25.57 | 3.80±1.08[b1] |

**Key**: DF, Dry feces; WF, Wet feces; NC, Negative control (DMSO); PC, Positive control (Loperamide 3 mg/kg); CK100, *C. kinondoensis* 100 mg/kg; CK200, *C. kinondoensis* 200 mg/kg, CK400, *C. kinondoensis* 400 mg/kg

[a] Compared with Negative control, [b] Compared with Positive control, [c] Compared with 100 mg/kg of *C. kinondoensis* extract. [d] Compared with 200 mg/kg of *C. kinondoensis* extract. [e] Compared with 400 mg/kg of *C. kinondoensis* extract. [1] P < .01, [2] P < .05, [3] P < .001

(NC), which received distilled water, exhibited frequent defecation and high wet fecal output, confirming the strong diarrheal effect of castor oil. The administration of loperamide (positive control group) significantly reduced diarrhea, as indicated by a 72.41% inhibition of defecation and a 71.43% inhibition of diarrhea, validating its effectiveness as a standard antidiarrheal drug.

Treatment with *C. kinondoensis* extract at 100 mg/kg, 200 mg/kg, and 400 mg/kg led to a gradual decrease in defecation frequency and wet fecal output, suggesting moderate antidiarrheal properties. The highest dose (400 mg/kg) exhibited the greatest reduction in diarrhea (25.57% inhibition of diarrhea and 20.69% inhibition of defecation), followed by 200 mg/kg (19.05% inhibition of diarrhea, 17.24% inhibition of defecation), and 100 mg/kg (14.29% inhibition of diarrhea, 10.34% inhibition of defecation). These findings indicate that *C. kinondoensis* may contain bioactive compounds that influence intestinal motility and secretion, although the overall inhibitory effect remained lower than that of loperamide. The results are visualized on Fig 1 above.

Despite the observed dose-dependent response (Fig 2), the antidiarrheal effect of *C. kinondoensis* was milder compared to loperamide, suggesting partial inhibition of diarrhea rather than complete suppression. This indicates that while the plant extract possesses potential therapeutic value, further investigation is needed to determine its mechanism of action, optimal dosage, and long-term safety profile. Future studies should explore whether combining the extract with other antidiarrheal agents could enhance its efficacy and provide a synergistic effect in managing diarrhea. Additionally, phytochemical characterization of the extract could help identify the active compounds responsible for its pharmacological effects, further supporting its traditional use in treating diarrhea.

**Antimotility test.**

The antimotility activity of *Croton kinondoensis* extract was assessed using the charcoal meal transit test, where a reduction in intestinal motility indicates a potential antidiarrheal effect. The negative control group (NC), which received distilled water, showed the highest peristaltic index (PI) of 54.1%, confirming the expected rapid intestinal transit induced by castor oil. The positive control group (PC), treated with loperamide (3 mg/kg), demonstrated a significant reduction in intestinal motility, with a peristaltic index of 36.0% and 33.43% inhibition of diarrhea, affirming its strong antimotility effect. (Table 3)

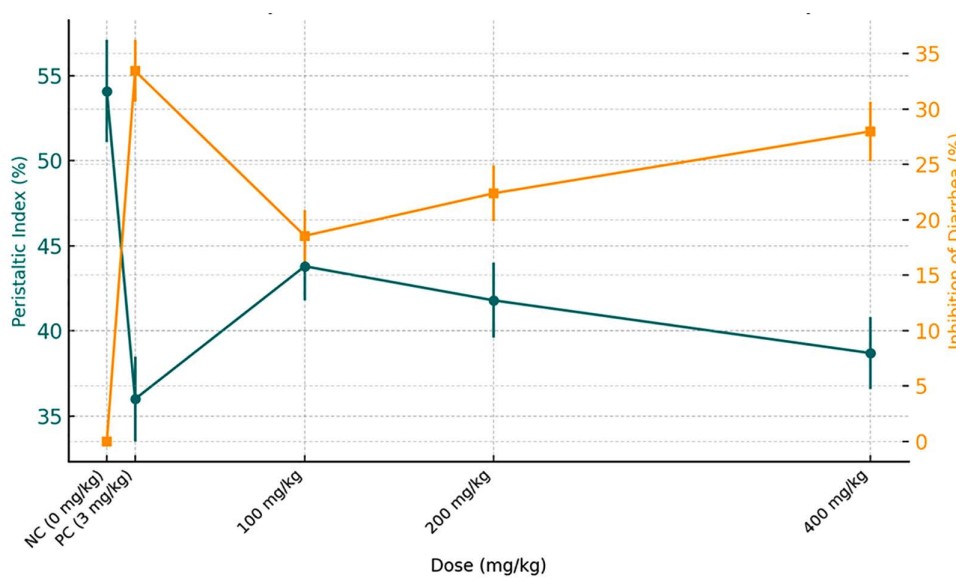

**Fig 1. Effect of *Croton kinondoensis* extract on Castor Oil-induced Diarrhea.**

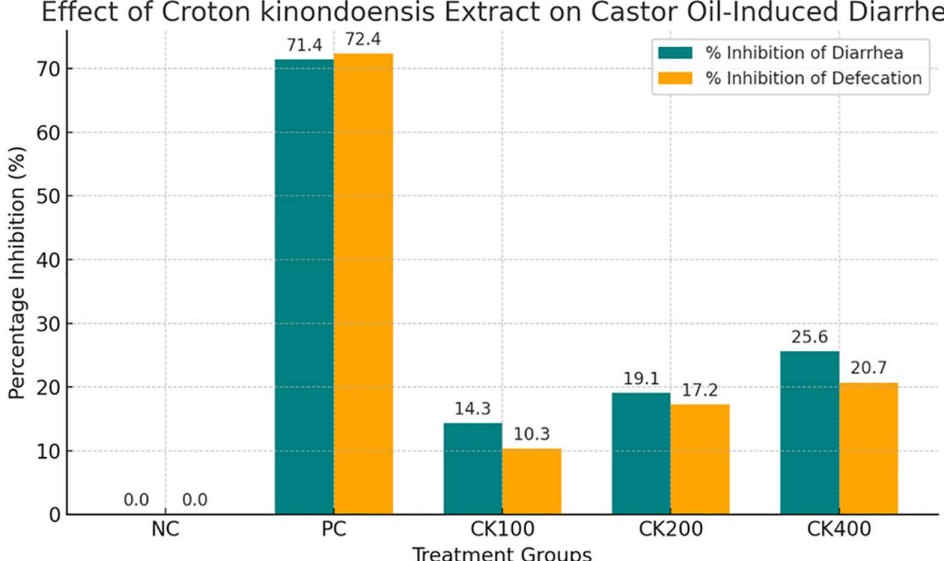

**Fig 2. Dose-Dependent effect of *Croton kinondoensis* on Castor Oil-Induced Diarrhea.**

**Table 3. Effect of Croton kinondoensis extract on antimotility test in mice.**

|  | Total length of the intestine | Distance travelled by charcoal | % Peristaltic Index (PI) | % Inhibition of Peristaltic Index (PI) |
|---|---|---|---|---|
| **NC** | 50.5±2.94 | 27.2±3.90 | 54.1±9.77 | ---- |
| **PC** | 49.3±2.87 | 17.7±1.0[a3c2] | 36.0±4.2[a3] | 33.43 |
| **CK100** | 49.8±1.6 | 21.8±1.94[a2b2] | 43.8±3.95[a2] | 18.56 |
| **CK200** | 49.5±1.52 | 20.7±1.63[a3] | 41.8±3.3[a2] | 22.39 |
| **CK400** | 50.3±1.86 | 19.5±2.43[a3] | 38.7±3.99[a3] | 27.98 |

**Key**: NC, Negative control (DMSO); PC, Positive control (loperamide 5 mg/kg); CK100, *C. kinondoensis* 100 mg/kg; CK200, *C. kinondoensis* 200 mg/kg, CK400, *C. kinondoensis* 400 mg/kg

[a]Compared with Negative control, [b]Compared with Positive control, loperamide 5 mg/kg, [c]Compared with 100 mg/kg of *C. kinondoensis* extract; [d] Compared with 200 mg/kg of *C. kinondoensis* extract; [e]Compared with 400 mg/kg of *C. kinondoensis* extract. [1] $P < .01$; [2] $P < .05$; [3]$P < .001$

Mice treated with *C. kinondoensis* extract exhibited a dose-dependent reduction in intestinal transit, supporting its antimotility activity. The 400 mg/kg dose showed the greatest inhibition of peristalsis (27.98%), with a peristaltic index of 38.7%, approaching the effect observed with loperamide. The 200 mg/kg dose also produced a moderate inhibition (22.39%), with a peristaltic index of 41.8%, whereas the 100 mg/kg dose exhibited the lowest inhibition (18.56%), with a peristaltic index of 43.8%.

These results suggest that *C. kinondoensis* possesses dose-dependent antimotility effects, likely contributing to its traditional use in treating diarrhea. However, while the highest dose (400 mg/kg) approached the effect of loperamide, the extract did not completely inhibit intestinal motility, implying a milder effect compared to the standard drug. Further studies are necessary to isolate the bioactive compounds responsible for the observed activity and to evaluate their precise mechanism of action in intestinal motility regulation. The results are highlighted in Figs 3 and 4 below:

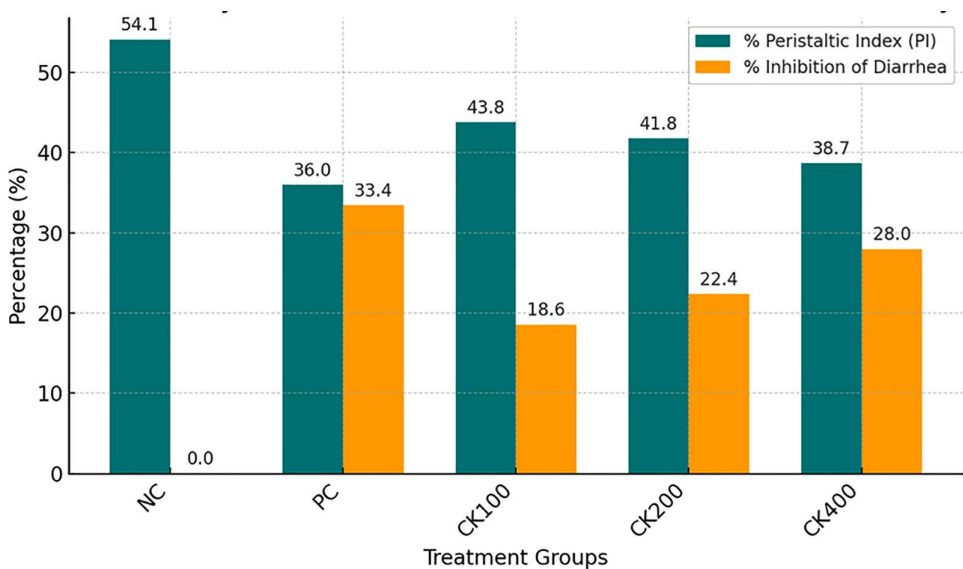

**Fig 3. Antimotility effects of *Croton kinondoensis* on intestinal motility.**

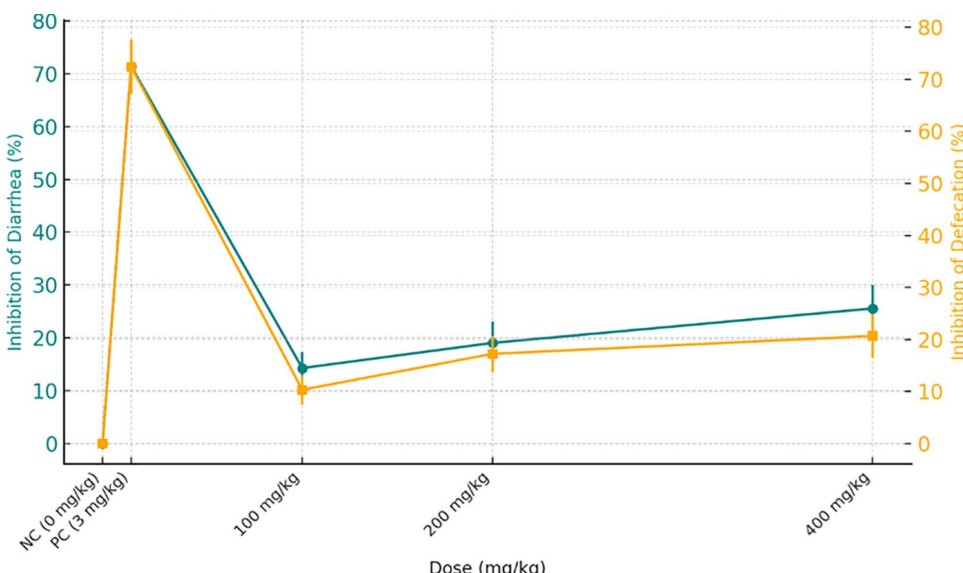

**Fig 4. Dose-dependent effect of Croton kinondoensis on antimotility test.**

## Discussion

Diarrhea remains a significant public health concern, particularly in developing regions where access to medical care and pharmaceutical treatments is often limited [19,20]. The current study evaluated the antidiarrheal and antimotility effects of *Croton kinondoensis* extract using Swiss albino mice, and the results indicate that the plant possesses moderate yet significant activity in alleviating diarrhea. The findings were assessed through phytochemical screening, acute toxicity testing,

castor oil-induced diarrhea model, and antimotility testing, with comparisons drawn against standard loperamide treatment and existing literature.

## Phytochemical Composition and Potential Mechanisms of Action

Phytochemical screening revealed the presence of flavonoids, phenolics, alkaloids, and tannins in *C. kinondoensis* extract. These secondary metabolites have been widely recognized for their antidiarrheal and spasmolytic properties. Studies have shown that flavonoids exert antidiarrheal effects by inhibiting intestinal motility and fluid secretion [21–23] Similarly, tannins enhance water absorption in the intestines and reduce peristalsis through protein precipitation and enzyme inhibition [24]. Alkaloids, on the other hand, exhibit spasmolytic activity, thereby slowing gut motility and decreasing diarrhea severity [25]. The presence of these bioactive compounds in *C. kinondoensis* suggests that its antidiarrheal effects are mediated through multiple mechanisms, including inhibition of intestinal motility, reduction of intestinal fluid secretion, and enhancement of mucosal protection.

## Acute Toxicity and Safety Profile

The acute toxicity test demonstrated that *C. kinondoensis* extract was well tolerated at a dose of 2000 mg/kg, with no observed mortality or significant behavioral changes over 14 days. According to the Organization for Economic Cooperation and Development (OECD) guidelines (2022), substances that exhibit no mortality at 2000 mg/kg can be considered relatively safe for therapeutic use. This finding aligns with previous reports on the safety of other Croton species used in traditional medicine [6]. There is a possibility that These constituents are likely responsible for the observed in vivo antidiarrheal activity of the extract [26,27]. However, chronic toxicity studies should be conducted to further assess the long-term effects of the plant extract.

## Castor oil-induced diarrhea model

The castor oil-induced diarrhea model is a widely accepted method for evaluating antidiarrheal efficacy, as ricinoleic acid, [28,29] the active metabolite of castor oil, stimulates intestinal motility, electrolyte secretion, and fluid accumulation, leading to watery stools [30–32]. In the present study, *C. kinondoensis* extract significantly reduced the number of wet feces and total fecal output in a dose-dependent manner, confirming its antidiarrheal potential.

At 400 mg/kg, the extract inhibited diarrhea by 25.57%, whereas loperamide (3 mg/kg) inhibited diarrhea by 71.43%. These findings suggest that *C. kinondoensis* has moderate antidiarrheal effects, though it was not as potent as loperamide, which is a well-established opioid receptor agonist that inhibits peristalsis and intestinal secretion [33]. A dose-dependent effect was evident, with 100 mg/kg and 200 mg/kg inhibiting diarrhea by 14.29% and 19.05%, respectively.

These results are consistent with previous studies on medicinal plants with antidiarrheal properties. A study by [16,34] on *Croton macrostachyus* and *Croton zambesicus* found significant inhibition of castor oil-induced diarrhea, similar to the present findings. Additionally, Gudeta et al reported that *Vernonia amygdalina* extracts reduced castor oil-induced diarrhea in rats, showing a comparable dose-dependent effect, reinforcing the traditional use of medicinal plants in gastrointestinal disorders [35].

The antidiarrheal effects of *C. kinondoensis* may be attributed to its phytochemical constituents, particularly flavonoids and tannins, which are well known for their ability to inhibit prostaglandin-induced intestinal secretion and motility [21]. Studies on *Guiera senegalensis* [36] and *Psidium guajava* [37] have similarly shown that flavonoid-rich extracts reduce stool frequency and improve intestinal water reabsorption, supporting the findings of the present study.

Furthermore, the ability of *C. kinondoensis* to reduce total fecal output and delay the onset of diarrhea suggests a potential mechanism involving modulation of intestinal motility and enhanced fluid reabsorption, as has been demonstrated in other traditional antidiarrheal plants. Similar findings were reported for *Punica granatum*, where its extract exhibited significant inhibition of diarrhea and slowed intestinal transit [38]

The present study, therefore, adds to the growing body of evidence supporting the antidiarrheal effects of medicinal plants, confirming that *C. kinondoensis* exhibits moderate yet significant efficacy. However, further research is needed to optimize extraction methods and identify the active compounds responsible for its effects.

Although the castor oil-induced enteropooling test is a standard model for assessing intestinal fluid secretion, it was not included in this study. Future investigations are encouraged to incorporate this method to evaluate potential antisecretory mechanisms of *C. kinondoensis* extract. Future investigations should also focus on synergistic effects when combined with conventional therapies to determine its potential for clinical application.

### Antimotility effects of *C. kinondoensis*..

The charcoal meal transit test is a well-established model for evaluating gastrointestinal motility, allowing for the assessment of potential antimotility effects of medicinal plant extracts. The results of the present study demonstrated that *C. kinondoensis* extract significantly reduced intestinal transit of the charcoal meal, exhibiting dose-dependent inhibition of peristalsis. At 400 mg/kg, the extract inhibited peristalsis by 27.98%, a value approaching that of Atropine (33.43%), a standard antidiarrheal agent. The 100 mg/kg and 200 mg/kg doses showed lower inhibition (18.56% and 22.39%, respectively), reinforcing the dose-dependent nature of its activity.

Loperamide exerts its antimotility effect by activating opioid receptors in the gastrointestinal tract, leading to reduced smooth muscle contractions and slowed intestinal transit [39]. While *C. kinondoensis* was less effective than loperamide, its significant inhibition of peristalsis suggests that it may share a similar mechanism of action. Potential pathways include inhibition of calcium channels, modulation of acetylcholine and serotonin release, or interactions with opioid receptors in the enteric nervous system.

These findings are in agreement with previous studies on other Croton species, which have also demonstrated antimotility activity. For example, *Croton zambesicus* and *Croton tiglium* have been reported to decrease peristaltic movement in animal models, confirming the presence of bioactive compounds with smooth muscle-relaxing properties [34]. Similar results have been observed in *Guiera senegalensis*, which significantly slowed intestinal transit, suggesting an interaction with the autonomic control of gut motility [40]. Moreover, *Psidium guajava* and *Punica granatum* extracts have shown strong antimotility effects, likely due to the presence of flavonoids and tannins, which have been implicated in smooth muscle relaxation [37,38].

The dose-dependent inhibition of peristalsis observed in this study further supports the antimotility potential of *C. kinondoensis*, suggesting that its bioactive compounds could delay stool passage and reduce the frequency of bowel movements. This aligns with previous research on medicinal plants traditionally used to manage diarrhea, reinforcing their role in modulating intestinal motility and fluid absorption. However, further pharmacological studies are needed to identify the specific active compounds responsible for the observed effects, as well as to investigate their mechanisms of action at the molecular level.

## Conclusion

The study demonstrated that *C. kinondoensis* extract possesses moderate antidiarrheal and antimotility properties, supporting its traditional use for managing diarrhea and abdominal cramps. This classification of the extract's antidiarrheal activity as moderate is based on established criteria, where an inhibition between 20–40% is considered moderate, 40–60% strong, and above 60% very strong [15,16]. The 25.57% inhibition observed at the highest tested dose therefore supports this interpretation.

Although its efficacy was lower than loperamide, the extract demonstrated significant potential as a natural remedy for diarrhea. The observed dose-dependent inhibition of diarrhea and intestinal motility is likely mediated by its flavonoid and tannin content, which may act via prostaglandin inhibition or smooth muscle relaxation However, further research is necessary to isolate the bioactive compounds, elucidate the mechanisms of action, and validate its clinical applicability.

## Supporting information

**S1 Data. Abdullahi Noor PROJECT RESULTS.**
(XLSX)

## Author contributions

**Conceptualization:** Ermias Mergai Terefe.

**Data curation:** Abdullahi Ibrahim Osman Noor.

**Formal analysis:** Abdullahi Ibrahim Osman Noor.

**Investigation:** Abdullahi Ibrahim Osman Noor.

**Methodology:** Abdullahi Ibrahim Osman Noor.

**Resources:** Ermias Mergai Terefe.

**Supervision:** Ermias Mergai Terefe.

**Validation:** Ermias Mergai Terefe.

**Visualization:** Abdullahi Ibrahim Osman Noor.

**Writing – original draft:** Abdullahi Ibrahim Osman Noor.

**Writing – review & editing:** Abdullahi Ibrahim Osman Noor.

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
