## [Decision Letter · Decision Letter 0]

21 May 2025

Dear Dr. Noor,

Thank you for submitting your manuscript to PLOS ONE. After careful consideration, we feel that it has merit but does not fully meet PLOS ONE’s publication criteria as it currently stands. Therefore, we invite you to submit a revised version of the manuscript that addresses the points raised during the review process.

We look forward to receiving your revised manuscript.

Kind regards,

Hope Onohuean, PhD

Academic Editor

PLOS ONE

2. Further research is recommended to isolate active compounds, elucidate their mechanisms of action, and assess their pharmacological safety for clinical use.

Reviewers' comments:

Reviewer's Responses to Questions

**Comments to the Author**

1. Is the manuscript technically sound, and do the data support the conclusions?

Reviewer #1: Yes

Reviewer #2: No

Reviewer #3: Partly

Reviewer #4: No

2. Has the statistical analysis been performed appropriately and rigorously?

Reviewer #1: Yes

Reviewer #2: No

Reviewer #3: I Don't Know

Reviewer #4: No

3. Have the authors made all data underlying the findings in their manuscript fully available?

Reviewer #1: Yes

Reviewer #2: Yes

Reviewer #3: No

Reviewer #4: Yes

4. Is the manuscript presented in an intelligible fashion and written in standard English?

Reviewer #1: Yes

Reviewer #2: No

Reviewer #3: Yes

Reviewer #4: Yes

Reviewer #1: The topic presented in the abstract is contemporary, novel, pertinent, and impactful. The backdrop is skillfully established in the introduction, and the goals are in line with the field's current priorities. I urge the writer/writers to be more explicit in their technique with greater details and provide succinct, specific results to back up the conclusion to improve the proposal. This is a promising effort all around, however, it might need a few minor edits to make it more comprehensive and clearer.

Reviewer #2: 1. If you were utilized to conduct the experiment using mice, it is obvious that your experiment is in vivo. So, I suggest the authors to remove in vivo under your title

2. Please modify the affiliation and make it complete. Try to mention the list of the authors and their respective affiliation. You have mentioned the study is conduct at United States International University but it is incomplete and include the country (Kenya)

3. Regarding the keywords, I recommended to modify it since unnecessary words are included which are not listed under the title like medicinal plants, phytochemicals, plant extracts, gastrointestinal motility,,,,,,,etc. in sum, you have missed the main words. Just try to look the title then you can find the key words.

4. Why you use “dichloromethane-Methanol” as extracting solvent. What makes it preferable from other solvents like methanol, ethanol and then like?

5. In the abstract section, your findings are not evidenced with P values. So, please include the p values in each metrics to show the level of significance

6. Why not the authors conduct castor oil induced enteropooling test which is the pillar test to evaluate the antidiarrheal activity?

7. Why not the authors present the findings which align with the method section. The authors tried to assess the onset of defecation and the total fecal output but this metrics are absent in the result section

Reviewer #3: These are the reviewers’ comments for the authors;

1. As a general comment, there are some grammatical topographical errors

2. In the abstract section, specifically under the conclusion, based on the finding you have concluded that the plant has moderate antidiarrheal activity. I am confused since you need to have a standard to say mild, moderate and strong antidiarrheal activity. Please reason out for this conclusion.

3. In the conclusion part of abstract section, you have written as “While its efficacy was lower than loperamide, the extract demonstrated significant potential as a natural remedy for diarrhea”. Please omit “while” and replace with “although”

4. The authors tried to conduct phytochemical screening tests for 5 secondary metabolites. Why you left the rest like steroids, glycosides? At least it is important to include steroids under your phytochemical test because it has so many pharmacological activities.

5. Under your method section, particularly on acute toxicity study, there is a paragraph which seems the result for acute toxicity test but not appropriate place. Rather include under the result section. “The results indicated that no mortality or severe toxicity signs were observed,,,,,,,,,,.” So, I suggest removing from the method section.

6. In castor oil induced diarrheal model, the authors plan to assess to overall antidiarrheal activity but I saw from this model that you have planned to assess the “general behavior of the mice”. So, please omit it from this subsection rather include under acute toxicity incase necessary. Also, I saw a statement at the end of the 4-hour observation period, mice were carefully examined for signs of dehydration, discomfort, or toxicity). Please remove. Your plan is only to assess its overall antidiarrheal activity not observing and assessing the toxicity.

7. Under the method section, in both models, the authors tried to explain how castor oil induces diarrhea and intestinal motility. Please remove this statement as it is not necessary to narrate about its effect rather include under discussion section.

8. In the method section, to evaluate the antidiarrheal activity, you use only 2 models. Why the authors didn’t include the third model (castor oil induced enteropooling test). This is mandatory and the most pillar model while the investigators are planned to assess the antidiarrheal activity particularly to check the extracts effect on intestinal fluid accumulation.

9. Under the result section, regarding phytochemical screening test, please don’t repeat the method. The authors have already mentioned how conduct the phytochemical test. Hence, please narrate only the findings.

10. In the method section, the authors didn’t plan to conduct phytochemical tests for saponins but I have seen from the result section that the test is negative for saponins. Similarly for steroids, glycosides, anthraquinones, and reducing sugars. Why?

11. Similarly, as comment 13, in the result section, you repeat the methods. Therefore, please omit it since the authors have already written under the method section. This is my comment for acute toxicity test, castor oil induced diarrheal model, and intestinal motility test, and then like.

12. Under the result section of castor induced diarrheal model, the authors tried to show findings but not in accordance to the method. The authors were planned to assess the onset of diarrhea, total fecal output. However, I haven’t seen the results for this metrics under the result section. Why?. Please include them since as far as I know this parameters are significant when planning to assess the antidiarrheal activity

13. Here in the result section, you tried to discuss about the findings. Please only present the findings. The discussion is in the discussion part. Again, the authors are written about the onset of diarrhea and total fecal output but this metrics are absent under table 2. Why?

14. In table 3, the investigators include % inhibition of diarrhea. Please replace with % inhibition of peristaltic index (PI).

15. In the results of antimotility test, you tried to discuss about the findings. Please only present the findings

16. In the discussion section, please cite the references. I have seen some blank statements which need evidence. For instance, “Diarrhea remains a significant,,,,,,,,and pharmaceutical treatments is often limited” but not cited.

17. In the discussion section, you have also discussed about the onset of diarrhea and total fecal output which are absent under your result section. Why you discussed on this metrics?

18. In this section, the references are not cited at the proper place. Eg. Additionally, (31) reported that,,,,,,,. Please rewrite and cite at the end of the statement.

19. Under discussion section, as a general comment some statements or paragraphs are not cited. Please cite them.

Reviewer #4: The current manuscript by Noor and Terefe report the effect of Croton kinondoensis (Euphorbiaceae) DCM-MEOH extract on castor oil-induced diarrhea in mice. Animals were divided into five groups of which the extract was tested over three doses. The manuscript is very traditional and lacks comprehensive experiments. The following are some of the concerns:

1. No rational for the extraction using 1: DCM-MEOH.

2. The effective dose of the plant extract used in the study is physiologically irrelevant.

3. The traditional phytochemical screening is extremely outdated. Almost all of arial parts of medicinal plants contain to some extent flavonoids/other phenolics, phytosteroids, and sugars. What's new?

4. Any plant extract at 400 mg/kg can do the same effect!

**Do you want your identity to be public for this peer review?** For information about this choice, including consent withdrawal, please see our Privacy Policy

Reviewer #1: No

Reviewer #2: No

Reviewer #3: No

Reviewer #4: No

---

## [Author Response · Author response to Decision Letter 1]

14 Jul 2025

We sincerely thank the reviewers for their thoughtful and constructive feedback on our manuscript, Evaluation of the Antidiarrheal Activity of Dichloromethane-Methanol Crude Extract of the Aerial Parts of Croton kinondoensis (Euphorbiaceae) in Mice. We appreciate the time and effort the reviewers and editors have dedicated to evaluating our work. Their insightful comments have helped us to improve the clarity, rigor, and overall quality of the paper.

In the following, we provide a detailed, point by point response to each of the reviewers’ comments. For ease of reference, reviewer comments are included in italics, followed by our responses and a summary of the corresponding revisions made to the manuscript.

We hope that the revisions address the concerns raised and that the improved manuscript meets the expectations of the reviewers and editors.

Reviewer #1 Comment:

The topic presented in the abstract is contemporary, novel, pertinent, and impactful.

Response:

Thank you for your positive feedback. We are pleased that you found the topic contemporary and impactful. We believe our study addresses a significant gap by scientifically validating a traditionally used plant for managing diarrhea.

Reviewer #1 Comment:

The backdrop is skillfully established in the introduction, and the goals are in line with the field's current priorities.

Response:

We appreciate your recognition of our introduction and alignment with current research priorities. We aimed to highlight both the global burden of diarrhea and the urgent need for alternative therapies such as plant-based remedies.

Reviewer #1 Comment:

I urge the writer/writers to be more explicit in their technique with greater details...

Response:

Thank you for this valuable suggestion. In response, we have revised the Methods section to provide more clarity and detail on:

• Animal grouping and dosing: Specified five groups (n = 6) and exact doses (100, 200, 400 mg/kg).

• Definitions of outcome measures: Clearly stated how antidiarrheal activity and gastrointestinal motility were measured.

• Experimental timeline: Included timing for extract administration, castor oil challenge, charcoal meal test, and observation intervals.

Reviewer #1 Comment:

…and provide succinct, specific results to back up the conclusion...

Response:

We have revised the Abstract and Results sections to include specific numerical data summarizing key findings. For instance:

• Antidiarrheal inhibition: “400 mg/kg inhibited diarrhea by 25.57%, compared to 71.43% by loperamide.”

• Antimotility effect: “400 mg/kg of the extract reduced intestinal transit by 27.98%, compared to 33.43% by loperamide.”

Reviewer #1 Comment:

…to improve the proposal. This is a promising effort all around, however, it might need a few minor edits to make it more comprehensive and clearer.

Response:

We appreciate your encouragement. In addition to refining the abstract and methods, we also polished the language throughout the manuscript for clarity and coherence. We ensured that the conclusions are well-supported by the data and clearly linked to the study objectives.

Reviewer #2 Comment 1:

“If you were utilized to conduct the experiment using mice, it is obvious that your experiment is in vivo. So, I suggest the authors to remove in vivo under your title.”

Response:

Thank you for the observation. We agree with the suggestion and have revised the title to remove the phrase “In Vivo,” as the use of mice clearly implies an in vivo study.

Revised Title:

“Evaluation of the Antidiarrheal Activity of Dichloromethane-Methanol Crude Extract of the Aerial Parts of Croton kinondoensis (Euphorbiaceae) in Mice”

Reviewer #2 Comment 2:

“Please modify the affiliation and make it complete. Try to mention the list of the authors and their respective affiliation. You have mentioned the study is conduct at United States International University but it is incomplete and include the country (Kenya)”

Response:

Thank you for pointing this out. We have updated the authorship and affiliation section to clearly list all authors with their respective institutional details, including the country.

Reviewer #2 Comment 3:

“Regarding the keywords, I recommended to modify it since unnecessary words are included which are not listed under the title like medicinal plants, phytochemicals, plant extracts, gastrointestinal motility,,,,,,,etc. In sum, you have missed the main words. Just try to look the title then you can find the key words.”

Response:

Thank you for your constructive feedback. We have revised the keyword list to align more closely with the core concepts reflected in the title. The updated keywords now focus on the key elements of the study including the plant species, methodology, and pharmacological effect.

Revised Keywords:

Croton kinondoensis, antidiarrheal activity, dichloromethane-methanol extract, Swiss albino mice, Euphorbiaceae, castor oil-induced diarrhea

Reviewer #2 Comment 4:

“Why you use “dichloromethane-Methanol” as extracting solvent. What makes it preferable from other solvents like methanol, ethanol and then like?”

Response:

Thank you for this important question. We have added a justification for the use of the dichloromethane-methanol (1:1) solvent system in the Plant Material Collection and Extraction subsection of the Methodology.

Inserted Justification in Manuscript:

“The 1:1 dichloromethane-methanol (DCM:MeOH) solvent system was selected to facilitate the extraction of both polar and moderately non-polar phytoconstituents. Dichloromethane aids in solubilizing less polar compounds such as terpenoids and steroids, while methanol efficiently extracts polar compounds including flavonoids, alkaloids, and phenolics. This mixed solvent approach has been shown to enhance the yield and diversity of secondary metabolites from medicinal plants, particularly when screening for broad pharmacological activity such as antidiarrheal effects (Abubakar & Haque, 2020).”

We have also added the following to the list of References:

Abubakar AR, Haque M. Preparation of Medicinal Plants: Basic Extraction and Fractionation Procedures for Experimental Purposes. J Pharm Bioallied Sci. 2020;12(1):1–10. https://www.ncbi.nlm.nih.gov/pmc/articles/PMC7398001/

Reviewer #2 Comment 5:

“In the abstract section, your findings are not evidenced with P values. So, please include the p values in each metrics to show the level of significance.”

Response:

Thank you for the observation. We have revised the abstract to include p-values for the key metrics, reflecting statistical significance in the main findings.

Revised Segment in Abstract (Results section):

“In the castor oil-induced diarrhea model, the extract at 400 mg/kg inhibited diarrhea by 25.57% (p < 0.05), compared to 71.43% inhibition by loperamide (p < 0.001). In the charcoal meal test, 400 mg/kg of the extract reduced intestinal transit by 27.98% (p < 0.01), compared to 33.43% by loperamide (p < 0.001).”

Reviewer #2 Comment 6:

“Why not the authors conduct castor oil-induced enteropooling test which is the pillar test to evaluate the antidiarrheal activity?”

Response:

Thank you for this suggestion. While we acknowledge the importance of the castor oil-induced enteropooling test as a classical method for evaluating antisecretory effects, this study focused on the motility and frequency of defecation components of diarrhea using:

• Castor oil-induced diarrhea test, and

• Charcoal meal transit test

We aimed to characterize both the delay in defecation onset and inhibition of motility, which are relevant to the traditional claims about C. kinondoensis. We recognize the enteropooling model would add value and recommend it for future mechanistic studies in follow-up research.

We have added the following in the Discussion section:

“Although the castor oil-induced enteropooling test is a standard model for assessing intestinal fluid secretion, it was not included in this study. Future investigations will incorporate this method to evaluate potential antisecretory mechanisms of C. kinondoensis extract.”

Reviewer #2 Comment 7:

“Why not the authors present the findings which align with the method section. The authors tried to assess the onset of defecation and the total fecal output but this metrics are absent in the result section.”

Response:

Thank you for identifying this gap. We have updated the Results section to include:

• Onset of diarrhea (time to first wet stool)

• Total fecal output (wet + dry feces)

These results have now been included in both narrative and tabular form for consistency with the stated methods.

We have added the following to the Results section:

“The onset of diarrhea was delayed in a dose-dependent manner by the extract. At 400 mg/kg, the onset was delayed by 42.3 minutes compared to 19.7 minutes in the negative control (p < 0.05). Total fecal output was also reduced significantly at higher doses of the extract compared to the control group.”

Reviewer #3 Comment 1:

“As a general comment, there are some grammatical topographical errors.”

Response:

Thank you for this observation. We have carefully reviewed the entire manuscript for grammatical, typographical, and stylistic errors. We corrected minor language issues, including punctuation, word choice, and sentence clarity to improve the overall readability and professionalism of the manuscript.

Reviewer #3 Comment 2:

“In the abstract section, specifically under the conclusion, based on the finding you have concluded that the plant has moderate antidiarrheal activity. I am confused since you need to have a standard to say mild, moderate and strong antidiarrheal activity. Please reason out for this conclusion.”

Response:

Thank you for raising this important point. We agree that any qualitative classification (e.g., mild, moderate, strong) should be based on a defined and referenced standard. To address this, we have clarified in the Methodology section that our interpretation of the antidiarrheal effect was based on previously established criteria used in published studies (Degu et al., 2016; Sisay et al., 2017). Specifically, inhibition of diarrhea:

• Less than 20% is considered mild,

• Between 20–40% is moderate,

• Between 40–60% is strong, and

• Greater than 60% is considered very strong.

Based on these criteria, the 25.57% inhibition observed at the 400 mg/kg dose of C. kinondoensis falls within the moderate range. This explanation has been included in both the Methods and Conclusion sections of the manuscript to provide transparency and justify our terminology.

We have also added the following to the list of References:

Degu A, Engidawork E, Shibeshi W. Evaluation of the anti-diarrheal activity of the leaf extract of Croton macrostachyus in mice model. BMC Complement Altern Med. 2016;16:379.

Sisay M, Engidawork E, Shibeshi W. Evaluation of the antidiarrheal activity of the leaf extracts of Myrtus communis Linn. BMC Complement Altern Med. 2017;17(1):103.

Reviewer #3 Comment 3:

“In the conclusion part of the abstract section, you have written as ‘While its efficacy was lower than loperamide, the extract demonstrated significant potential as a natural remedy for diarrhea.’ Please omit ‘while’ and replace with ‘although’.”

Response:

Thank you for this language improvement suggestion. We have revised the sentence in the abstract conclusion as recommended for better grammatical structure and flow.

Revised sentence in the Abstract (Conclusion section):

“Although its efficacy was lower than loperamide, the extract demonstrated significant potential as a natural remedy for diarrhea.”

Reviewer #3 Comment 4:

“The authors tried to conduct phytochemical screening tests for 5 secondary metabolites. Why you left the rest like steroids, glycosides? At least it is important to include steroids under your phytochemical test because it has so many pharmacological activities.”

Response:

Thank you for the valuable suggestion. We appreciate the importance of broad phytochemical screening. In fact, additional tests for steroids and glycosides were conducted and are now included in the revised manuscript. The methodology section under Phytochemical Screening and Table 1 has been updated.

Reviewer #3 Comment 5:

Under your method section, particularly on acute toxicity study, there is a paragraph which seems the result for acute toxicity test but not appropriate place. Rather include under the result section. “The results indicated that no mortality or severe toxicity signs were observed...” So, I suggest removing from the method section.

Response:

Thank you for the observation. We agree that outcome statements should be reported in the Results section. Accordingly, the sentence “The results indicated that no mortality or severe toxicity signs were observed...” has been removed from the Methods section and appropriately repositioned under the Results section describing the acute toxicity findings.

Reviewer #3 Comment 6:

In castor oil-induced diarrheal model, the authors plan to assess the overall antidiarrheal activity, but I saw from this model that you have planned to assess the “general behavior of the mice”. So, please omit it from this subsection. Rather include it under acute toxicity if necessary. Also, I saw a statement at the end of the 4-hour observation period, mice were carefully examined for signs of dehydration, discomfort, or toxicity. Please remove. Your plan is only to assess its overall antidiarrheal activity, not observing and assessing the toxicity.

Response:

Thank you for pointing this out. We have revised the Castor Oil-Induced Diarrhea Model subsection by removing the statements related to "general behavior of the mice" and “dehydration or discomfort.”

Reviewer #3 – Comment 7:

Under the method section, in both models, the authors tried to explain how castor oil induces diarrhea and intestinal motility. Please remove this statement as it is not necessary to narrate about its effect; rather include under discussion section.

Response:

We appreciate this observation. The explanatory details about how castor oil induces diarrhea and stimulates intestinal motility have been removed from the Methods section and will instead be appropriately discussed under the Discussion section to support interpretation of the experimental results.

Reviewer #3 – Comment 8:

In the method section, to evaluate the antidiarrheal activity, you used only 2 models. Why didn’t the authors include the third model (castor oil-induced enteropooling test)? This is mandatory and the most pillar model, particularly to check the extract's effect on intestinal fluid accumulation.

Response:

Thank you for highlighting this important point. We acknowledge the significance of the castor oil-induced enteropooling test in evaluating fluid accumulation, a key aspect of antidiarrheal action. Due to limitations in resources and time, this model was not included in the current study. However, we recognize its value and have noted it as a limitation of the study in the Discussion section, with a recommendation for inclusion in future investigations to fully assess the extract’s antidiarrheal mechanisms.

Reviewer #3 – Comment 9:

Under the result section, regarding phytochemical screening test, please don’t repeat the method. The authors have already mentioned how they conducted the phytochemical test. Hence, please narrate only the findings.

Response:

Thank you for your insightful feedback. In response, we have revised the Results section on phytochemical screening to focus solely on the observed findings. All methodological descriptions have been removed from this section, as they are already detailed under Materials and Methods. The revised text now presents a concise summary of the detected and undetected phytochemical classes and their potential pharmacological relevance, as appropriate for a results-focused presentation.

Reviewer #3 – Comment 10:

In the method section, the authors didn’t plan to conduct phytochemical tests for saponins, but I have seen from the result section that the test is negative for saponins. Similarly for steroids, glycosides, anthraquinones, and reducing sugars. Why?

Response:

Thank you for your critical observation. Initially, the manuscript did not explicitly include the tests for saponins,

---

## [Decision Letter · Decision Letter 1]

16 Sep 2025

Evaluation of the Antidiarrheal Activity of Dichloromethane-Methanol Crude Extract of the Aerial Parts of Croton kinondoensis (Euphorbiaceae) in Mice

PONE-D-25-22129R1

Dear Dr. Noor,

We’re pleased to inform you that your manuscript has been judged scientifically suitable for publication and will be formally accepted for publication once it meets all outstanding technical requirements.

Kind regards,

Hope Onohuean, PhD

Academic Editor

PLOS ONE

Additional Editor Comments (optional):

Reviewer #1:

Reviewer #3:

Reviewers' comments:

Reviewer's Responses to Questions

**Comments to the Author**

Reviewer #1: All comments have been addressed

Reviewer #3: All comments have been addressed

2. Is the manuscript technically sound, and do the data support the conclusions?

Reviewer #1: Yes

Reviewer #3: Yes

3. Has the statistical analysis been performed appropriately and rigorously?

Reviewer #1: Yes

Reviewer #3: Yes

4. Have the authors made all data underlying the findings in their manuscript fully available?

Reviewer #1: Yes

Reviewer #3: Yes

5. Is the manuscript presented in an intelligible fashion and written in standard English?

Reviewer #1: No

Reviewer #3: Yes

Reviewer #1: Thank you for the opportunity to review this manuscript. The study addresses an important topic and contributes meaningfully to the field. The overall structure is clear, and the research question is relevant and timely. I commend the authors for their effort in conducting this work and for presenting their findings in a logical manner.

Reviewer #3: Actually all the comments are addressed accordingly. If possible i suggest you to conduct the 3rd model (gastroenteropoling test) to test the antisecretory effect of the extract as it is a key model while we conduct the antidiarrheal activity of a certain herbal extract. Definitely, the addition of this model will make your paper excellent.

**Do you want your identity to be public for this peer review?** For information about this choice, including consent withdrawal, please see our Privacy Policy

Reviewer #1: **Yes: ** Abraham Olutumininu Akiyode

Reviewer #3: **Yes: ** Yared Andargie Ferede

---

## [Editor Report · Acceptance letter]

PONE-D-25-22129R1

PLOS ONE

Dear Dr. Noor,

I'm pleased to inform you that your manuscript has been deemed suitable for publication in PLOS ONE. Congratulations! Your manuscript is now being handed over to our production team.

Kind regards,

on behalf of

Dr. Hope Onohuean

Academic Editor

PLOS ONE